# Quality of basic emergency obstetric and newborn care services from patients' perspective in selected public health centers in Addis Ababa, Ethiopia 2022: A cross-sectional study

**Willi Bahre** [1]*, **Achamyelesh Tadele**[2], **Finot Debebe**[2]

**1** Department of Nursing, College of Medicine and Health Sciences, Adigrat University, Adigrat, Tigray, Ethiopia, **2** Department of Emergency Medicine, College of Health Sciences, Addis Ababa University, Addis Ababa, Ethiopia

* willibahre21@gmail.com

## Abstract

### Background

The majority of maternal and neonatal deaths occur within the first 24 hours of birth. To minimize maternal as well as neonatal morbidity and mortality, it is important to supply quality Basic Emergency Obstetric and Newborn Care. Basic emergency obstetric and newborn care services prevent immediate obstetric problems. There have been studies in Ethiopia that have looked at the availability of EmONC services. However, from the clients' perspective and experience, there is insufficient knowledge of quality BEmONC services.

### Objective

To assess the quality of basic emergency obstetric and newborn care (BEmONC) services and associated factors from the perspective of mothers in selected public health centers in Addis Ababa, Ethiopia, 2022.

### Methods

A facility-based cross-sectional study was used among mothers receiving at least one of the signal functions of BEmONC services. A total of 377 mothers were enrolled. Eleven public health centers, one from each of the 11 sub-cities, were selected by simple random sampling. Respondents were chosen by a systematic random sampling method. A structured questionnaire from Open Data Kit version 2022.1.2 was used. Finally, it was exported to SPSS version 26 for analysis. Bivariate analysis at a P-value of 0.25 and multivariable analysis at a P-value of 0.05 were applied.

**Data availability statement:** All relevant data are within the paper and its Supporting Information files.

**Funding:** The author(s) received no specific funding for this work.

**Competing interests:** The authors declare that they have no competing interests.

**Abbreviations:** ANC, Antenatal care; AOR, Adjusted Odds Ratio; BEmONC, Basic Emergency Obstetric and Newborn Care; CEmONC, Comprehensive Emergency Obstetric and Newborn Care; EDHS, Ethiopian Demographic and Health Survey; EmONC, Emergency Obstetric and Newborn Care; ODK, Open Data Kit; SPSS, Statistical Package for Social Sciences; UNFPA, United Nations Population Fund; UNICEF, United Nations Children's Fund; VIF, Variance Inflation Factor; WHO, World Health Organization.

## Results

The overall quality of BEmONC services from the mothers' perspective was 56.9%. Mothers who paid for services had lower odds of rating the quality as good compared to those who received services for free (AOR = 0.564; 95% CI: 0.327–0.971). Additionally, mothers aged 20 to 24 years had a lower likelihood of viewing the quality as good compared to those older than 35 years (AOR = 0.362; 95% CI: 0.157–0.837). However, mothers who were accompanied by relatives had significantly higher odds of rating the quality as good than those who were alone (AOR = 18.557; 95% CI: 3.844–89.588). Regarding monthly income, respondents with an average monthly income of less than 1,500 ETB had higher odds of rating the quality as good compared to those earning more than 6,000 ETB (AOR = 2.429; 95% CI: 1.026–5.753).

## Conclusion and recommendation

The total quality of BEmONC services from the perspective of mothers was suboptimal. It was predicted by age, monthly income, presence of a companion, and payment. This study strongly recommends that more should be done to ensure that the services given are more client-centered.

## Introduction

Maternal death is described as a mother's death in pregnancy, childbirth, or the early forty-two days after birth, no matter the pregnancy's duration or location, of any etiology linked with or exacerbated by childbirth or its treatment but not related to incidental or unintentional events [1]. During pregnancy, labor, delivery, and the postpartum period, global estimates suggest that 15% of all expected deliveries result in life-threatening complications [2].

According to the World Health Organization (WHO), 580,000 women of reproductive age die each year as a result of pregnancy problems, with Sub-Saharan Africa accounting for the majority of these deaths. With 686 deaths per 100,000 live births, the region has one of the highest maternal death rates in the world. Ethiopia is among the countries with the world's highest maternal death rates. The maternal death rate in Ethiopia was predicted to be 412 per 100,000 live births, according to the Ethiopian demographic and health survey (EDHS 2016) [3]. Poor delivery care caused some of these deaths.

To reduce maternal and neonatal mortality, the World Health Organization (WHO) created and developed Emergency Obstetric and Newborn Care (EmONC) [4]. Emergency obstetric and newborn care is a vital sequence of life-saving tasks that can save the lives of mothers who are experiencing obstetric difficulties [5]. Basic Emergency Obstetric and Newborn Care (BEmONC) can prevent intrapartum neonatal and maternal deaths by up to 40% [6].

Basic Emergency Obstetric and Newborn Care (BEmONC) services encompass the giving of parenteral antibiotics, uterotonic medicine, anticonvulsants, manual placenta removal, removal of residual tissue products, guided vaginal birth, and neonatal resuscitation, which are the 7 signal functions.

Ethiopia has created and developed a thorough national guideline to ensure that quality institutional delivery services are provided at all levels of health facilities. Input, process, and output (satisfaction) are all clearly identified in the Donabedian model guideline as the three quality components. But only some studies have attempted to assess the quality of emergency obstetric and newborn care services in health institutions from the perspective of patients using all three components of the Donabedian model [7–9].

Client satisfaction with healthcare professionals' services can be rated by taking into consideration clients' perceptions of maternal and neonatal health care services [10]. There have been studies in Ethiopia that have looked at the availability of EmONC services [8,11–13]. But, from the clients' perspective and experience, there is insufficient knowledge of quality BEmONC service [14].

As a result, the Donabedian model was used to assess the quality of basic emergency obstetric and newborn care services and factors affecting women's ratings for quality from the patients' perspective in the health centers of Addis Ababa in this study. This is going to aid in documenting the quality of EmONC services from the perspective of mothers, which is necessary for developing patient-centered basic emergency obstetric and newborn care (BEmONC) standards. It can be used as a baseline study and as a guide for conducting further studies.

## Materials and methods

### Study setting and study period

According to the 2019–2020 annual performance report, Addis Ababa has 41 hospitals (13 public and 28 NGO and private), 98 health centers, 122 health stations, 37 health posts, and 382 modern private clinics. Each of the public health centers serves an estimated 40,000 people. This study was conducted in the eleven public health centers in Addis Ababa, Ethiopia, which were selected by simple random sampling, one from each sub-city. All the selected eleven public health centers provide BEmONC services. A total of 771 mothers utilized Basic Emergency Obstetric and Newborn Care (BEmONC) services in the chosen eleven public health centers during the specific study period. This study was conducted from April 18, 2022, to May 19, 2022.

### Study design

A facility-based cross-sectional study was conducted among mothers receiving basic emergency obstetric and newborn care services.

### Population

All mothers in the reproductive age groups who had visited public health centers in Addis Ababa were the source population, while all mothers who gave birth in Addis Ababa's randomly selected public health centers during the data collection period were the study population for this study.

### Inclusion and exclusion criteria

Mothers were scheduled for discharge after getting at least one of the basic emergency obstetric and newborn care signal functions, as well as postnatal mothers within 42 days postpartum who attended postnatal follow-up at the same health center where they gave birth and received at least one of the seven signal functions, were included in this study. However, women who were referred to other health institutions, critically or mentally ill mothers, mothers in severe pain, and those unwilling to participate were excluded from this study.

### Sample size determination and sampling procedure

A single population proportion formula was used to calculate the sample size. Based on a prior study, it was found that 66.3 percent of BEmONC services were of good quality from the mothers' perspective [7], with a 95 percent confidence level, a desired degree of precision of 5 percent, and a 10 percent contingency for the non-response rate. Finally, with n = 343 and a 10% non-response rate, the total number of mothers needed for this study was 377.

This study included eleven public health centers in Addis Ababa that were chosen using a simple random selection technique from the 11 sub-cities, one from each sub-city. All the selected eleven public health centers provided Basic Emergency Obstetric and Newborn Care (BEmONC) services. A one-month pre-assessment survey of mothers receiving the services was used to predict the flow of mothers and proportionate the size accordingly.

An average of 761 one-month postnatal mothers received BEmONC services at the eleven public health centers in the month prior to the study. The sampling interval was calculated by dividing the average one-month postnatal mother by the total sample size, and this interval was used in all health centers to select study subjects (k = 2). A systematic random sampling technique was applied to obtain 377 study subjects. The sampling interval was calculated by dividing the average one-month postnatal mothers by the total sample size, and this interval was used in all health centers to select study subjects (k = 2). The first mother was selected randomly from those postnatal mothers during the first day of the data collection period.

## Data collection instruments and procedures

Five BSc and one MSc midwives were recruited to collect the data, and they were given training on basic data collection skills for two days and orientation on the Open Data Kit (ODK) application. The sample size was proportionally allocated to the randomly chosen health centers based on the predicted number of mothers attending throughout the data collection period. Then, from each of the randomly chosen public health centers, systematic random sampling was employed to pick study participants. Finally, mothers who gave birth and received at least one of the signal functions of basic emergency and newborn care services were interviewed.

Data were collected using a structured tool that was adapted from a similar study done in northern Ethiopia [7]. The English form of the questionnaire was first translated into Amharic. The questionnaire covered socio-economic data, obstetric characteristics, quality questions, and satisfaction questions. The ODK version 2022.1.2 software was used to collect the data, along with the Kobo Toolbox humanitarian response server to store the collected data. A total of 53 questions were asked.

## Operational definitions

**BEmONC:** Services involve 7 signal functions like providing parenteral antibiotics, uterotonic drugs, anticonvulsants, manual placenta removal, removal of residual tissue, guided vaginal birth, and neonatal resuscitation [15].
**CEmONC:** In addition to the seven signal functions of BEmONC services, those include two signal functions: cesarean section and blood transfusion [15].
**Signal functions:** WHO, UNICEF, and UNFPA identified a set of interventions that can be used to manage direct obstetric complications. These interventions are crucial in (EmONC) [16].
**Quality:** The degree to which health services for individuals and populations increase the likelihood of desired health outcomes in line with evidence-based professional knowledge [17].
**Magnitude of quality with service:** strongly agree (very satisfied)" and "agree (satisfied)" were categorized as agree (satisfied), while "strongly disagree (very dissatisfied)," "disagree (dissatisfied)," and "neutral" were categorized as disagree (dissatisfied). Neutral replies were classified as disagreeing (dissatisfied) because they could indicate a modest method of expressing dissatisfaction. This is likely due to the fact that since the interview took place in a health facility, women might be unwilling to share their unhappiness with the care they received [13].

### Level of quality score in mean and percentage

**Good quality:** The quality of BEmONC services from the mothers' perspective when the mothers scored greater than or equal to the mean of 104 (70%) of all the quality questions [7]
**Poor quality:** When the mothers scored below the mean of 104.19 (69.9%) of all the quality questions [7].
**Patient perspective (experience):** patient feedback on the course of getting care or intervention, including both objective facts and a subjective view of it. The factual component is useful for comparing what people claim they experienced with what an agreed-upon pathway or quality standard says should happen [18].

### Measurement of quality

**Donabedian's framework.** The Donabedian model is built on three quality components: input, process, and output/satisfaction. Input refers to the physical and institutional aspects of care settings, such as employees, facilities, and other material resources. Patients seeking care and professionals' treatments and suggestions are examples of clinical encounters that describe the process. These two components interact to change the outcome, which encompasses changes in health status, behavior, and health literacy of the patients and populations. These three variables can be used to make inferences regarding health care quality [10]. As a result, the prior study [7] applied this model to build the questionnaire that we have taken.

**Data quality assurance.** To ensure the quality of the data, the data collectors and supervisors were trained for 2 days on the basics of data collection skills and orientation to the ODK application. The questionnaire was designed on the Kobo Toolbox server carefully to allow optional and mandatory questions by skip logic to minimize missing data. The tool, which was first written in English, was translated into Amharic.

Before the actual data collection period, the questionnaire was checked for clarity, comprehensiveness, and internal consistency reliability using a Cronbach alpha ($\alpha = 0.946$) on 5% of the sample. Finally, possible modifications were made to the questions. The supervisor evaluated and validated the obtained data for completeness and consistency during data collection, and data collectors were immediately notified if the survey forms were incomplete or incorrectly filled in.

**Data processing and analysis.** Open Data Kit (ODK) version 2022.1.2 software was used to collect the data, along with the Kobo Toolbox server to store the collected data, and it was exported to SPSS version 26 for analysis. Before data export, it was checked for completeness, and frequency was run to rule out any missing values. Then the data was coded and revalued on SPSS, frequency distributions were run, and further cleansing and checking for missing values and errors were done.

The sample was described using descriptive statistics, and numerical data was reported as mean, standard deviation, proportion, or percent. A logistic regression model was used for both bivariate and multivariable analysis to identify determinants of the quality of BEmONC services among groups of independent variables. Independent variables with a p-value of < 0.25 that were biologically plausible and showed significant associations in the previous studies were included in the multivariable analysis to control for all possible confounders.

The Hosmer and Lemeshow statistics were used to check the goodness of fit of the model. The variance inflation factor (VIF) was used to assess multicollinearity. However, no multicollinearity was detected as the variance inflation factor was less than five. An adjusted odds ratio (AOR) with a 95% CI was estimated to assess the strength of associations, and statistical significance was declared at a p-value $\leq 0.05$. Results were presented using tables, figures, and texts.

**Ethical approval and consent to participate.** Ethical clearance was received from the departmental research and ethical review committee of the department of emergency medicine at Addis Ababa University. An official letter of permission from the department was submitted to the selected health centers. Informed consent was obtained from each participant before an interview. Throughout the study period, information was recorded anonymously, and confidentiality was ensured. All methods were performed in accordance with the Declaration of Helsinki.

## Results

### Socio-demographic/economic features

Out of 377 eligible mothers, 353 women who gave birth at the 11 public health centers found in Addis Ababa took part in the interview, which resulted in a response rate of 94%. A total of 153 (43.3%) of the women were aged between 25 and 29 years, with a mean of 28 (± 4.7); 339 (96%) of the women were from urban areas; 41 (11.6%) had not had any formal education; 206 (58.4%) of them were housewives; and 335 (94.9%) of them were married. The median monthly income was 4,500 ETB per month (Table 1).

### Obstetric characteristics of participants

From the total of 353 mothers, more than half (61.5%) were multigravida; the vast majority (97.7%) of the mothers had antenatal care (ANC) visits for this current pregnancy; and 339 (96%) of the mothers became pregnant by choice for this current pregnancy. Two hundred twenty-three (63.2%) participants waited less than 15 minutes, followed by 68 (19.3%) 15 to 30 minutes, 33 (9.3%) above 1 hour, and 29 (8.2%) 30 minutes to 1 hour before receiving any service or care. Twenty (5.7%) of the women had no companion (attendant) in the waiting room, labor unit, and postnatal unit during their stay. Spontaneous vaginal delivery was the dominant mode of delivery (96.3%), followed by 13 (3.7%) assisted vaginal deliveries, but there was no report of abortion. Two-thirds (75.6%) of the women did not pay for any of the services in the health centers during their stay (Table 2).

### Quality of basic emergency obstetric and newborn care services from the perspective of mothers

**Input.** From the 353 respondents, the overall quality of the input from the perspective of women, those scored greater than or equal to the mean of 15 (60%) or rated as good quality, was 174 (49.3%). The mothers expressed that the most contributing characteristic for rating the quality as good was the sanitation of the wards (70.5%), while the most contributing characteristic for rating the quality as poor was the availability of functional and clean shower and toilet rooms (38.2%) (Table 3).

**Process.** From the 353 respondents interviewed, the overall quality regarding the process from the perspective of women, those who scored greater than or equal to the mean of 57 (70%) or rated as good quality, was 216 (61.2%). The women expressed staff support in breastfeeding their babies immediately after birth as the most important aspect in rating them as good quality (90.7%). However, asking permission before applying any examination or procedure (48.4%) was the major contributing aspect to rating them as poor quality, followed by the availability of health providers (59.8%) (Table 4).

**Outcome (satisfaction).** Out of the 353 respondents, the overall quality of the outcome from the perspective of women, those who scored greater than or equal to the mean of 31 (70%) or rated as good quality, was 192 (54.4%). Mothers expressed that the most contributing characteristic for rating them as good quality was health professionals' respect for their

**Table 1. Socio- demographic/economic characteristics of participants at public health centers in Addis Ababa, Ethiopia, in 2022 (n = 353).**

| Characteristics | Frequency | Percent |
|---|---|---|
| **Age category** | | |
| 15–19 | 3 | 0.8 |
| 20–24 | 68 | 19.3 |
| 25–29 | 153 | 43.3 |
| 30–34 | 79 | 22.4 |
| >35 | 50 | 14.2 |
| **Residence** | | |
| Urban | 339 | 96 |
| Rural | 14 | 4 |
| **Educational level** | | |
| No formal education | 41 | 11.6 |
| Primary level | 140 | 39.7 |
| Secondary level | 104 | 29.5 |
| Diploma | 31 | 8.8 |
| Degree and above | 37 | 10.5 |
| **Occupation** | | |
| Governmental | 41 | 11.6 |
| Private | 84 | 23.8 |
| Daily laborer | 21 | 5.9 |
| House wife | 206 | 58.4 |
| Others | 1 | 0.3 |
| **Marital status** | | |
| Single | 4 | 1.1 |
| Married | 335 | 94.9 |
| Divorced | 9 | 2.5 |
| Widowed | 5 | 1.4 |
| **Husband's education** | | |
| No formal education | 21 | 5.9 |
| Primary level | 85 | 24.1 |
| Secondary level | 133 | 37.7 |
| Diploma | 30 | 8.5 |
| Degree and above | 66 | 18.7 |
| **Husband's occupation** | | |
| Governmental | 71 | 20.1 |
| Private | 187 | 53 |
| Daily laborer | 53 | 15 |
| Unemployed | 4 | 1.1 |
| Driver | 17 | 4.8 |
| Other | 3 | 0.8 |
| **Average monthly income category (ETB)** | | |
| 0–1500 | 40 | 11.3 |
| 1,501–3,000 | 100 | 28.3 |
| 3,001–4,500 | 43 | 12.2 |
| 4,501–6,000 | 66 | 18.7 |
| >6,000 | 104 | 29.5 |

**Table 2. Obstetric characteristics of participants at public health centers in Addis Ababa, Ethiopia, 2022 (n = 353).**

| Characteristics | Frequency | Percent |
|---|---|---|
| **Gravidity** | | |
| Primigravida | 136 | 38.5 |
| Multigravida | 217 | 61.5 |
| **ANC Visit** | | |
| Yes | 345 | 97.7 |
| No | 8 | 2.3 |
| **Desire of Current Pregnancy** | | |
| Wanted | 339 | 96 |
| Unwanted | 14 | 4 |
| **Type of Visit** | | |
| Planned (Direct) | 346 | 98 |
| Referred | 7 | 2 |
| **Mode of Transportation** | | |
| Ambulance | 49 | 13.9 |
| Public transportation | 228 | 64.6 |
| On foot | 44 | 12.5 |
| Ride | 23 | 6.4 |
| Other | 9 | 2.6 |
| **Waited Time to Receive service** | | |
| < 15 Minutes | 223 | 63.2 |
| 15 – 30 Minutes | 68 | 19.3 |
| 30 Minutes – 1 Hour | 29 | 8.2 |
| >1 Hours | 33 | 9.3 |
| **Presence of Companion During Stay** | | |
| Yes | 333 | 94.3 |
| No | 20 | 5.7 |
| **Mode of Delivery** | | |
| Spontaneous Vaginal Delivery | 340 | 96.3 |
| Assisted Vaginal Delivery | 13 | 3.7 |
| **Health Outcome of Mother after Delivery** | | |
| Normal | 312 | 88.4 |
| With Complication | 41 | 11.6 |
| **Birth Outcome of Newborn** | | |
| Alive | 352 | 99.7 |
| Still birth | 1 | 0.3 |
| **Health Problem of Newborn** | | |
| Yes | 41 | 11.6 |
| No | 312 | 88.4 |
| **Payment** | | |
| Yes | 86 | 24.4 |
| No | 267 | 75.6 |

privacy during their stay (73.1%), whereas the most common aspect contributing to rating the quality as poor was involving them in decision-making about themselves and their baby's condition (43.1%), followed by the number of health workers in the labor and delivery ward (51.3%) (Table 5).

**Table 3. Quality of BEmONC services regarding input at public health centers in Addis Ababa, Ethiopia, 2022 (n = 353).**

| Characteristics | Good F (%) | Poor F (%) |
|---|---|---|
| Availability of necessary equipment's | 211(59.8) | 142(40.2) |
| Adequate no of health providers | 183(51.8) | 170(48.2) |
| Sufficient bed, room, and space for laboring and delivering mothers | 209(59.2) | 144(40.8) |
| Sanitation of the wards | 249(70.5) | 104(29.5) |
| Functional and clean shower and toilet room | 135(38.2) | 218(61.8) |

**Table 4. Quality of BEmONC services regarding processes, at public health centers found in Addis Ababa city, Ethiopia, 2022 (n = 353).**

| Characteristics | Good | Poor |
|---|---|---|
| Respect and courtesy by the health providers | 237 (67.1) | 116 (32.9) |
| The environment where you were laboring was comfortable | 257 (72.8) | 96 (27.2) |
| Active follow-up on the progress of labor | 256 (72.5) | 97 (27.5) |
| Permission before applying any examination and procedures | 171 (48.4) | 182 (51.6) |
| Progress of labor explanation by using clear and local language | 221 (62.6) | 132 (37.4) |
| Similar advice or information given by staff members | 259 (73.4) | 94 (26.6) |
| Health workers spent enough time for examination | 246 (69.7) | 107 (30.3) |
| Verbally encouraged, reassured and praised | 252 (71.4) | 101 (28.6) |
| Received enough care and help during the course of labor | 228 (64.6) | 125 (35.4) |
| Competence and confidence of health providers | 230 (65.2) | 123 (34.8) |
| Privacy well-kept in the delivery room | 255 (72.2) | 98 (27.8) |
| Received adequate care and support during delivery | 226 (64) | 127 (36) |
| Availability of health providers | 211 (59.8) | 142 (40.2) |
| Breast-feeding assistance from the staff immediately after birth | 320 (90.7) | 33 (9.3) |
| Got counseling on how to take care of baby | 248 (70.3) | 105 (29.7) |
| Your baby received enough care and support | 235 (66.6) | 118 (33.4) |

**Table 5. Quality of BEmONC services regarding output at public health centers in Addis Ababa city, Ethiopia, 2022. (n = 353).**

| Characteristics | Good F (%) | Poor F (%) |
|---|---|---|
| Respected your personal culture and religion | 215 (60.9) | 138 (39.1) |
| Health care providers respect of your privacy during your stay | 258 (73.1) | 95 (26.9) |
| No of health care workers in labor and delivery ward | 181 (51.3) | 172 (48.7) |
| Health workers competency and confidence | 236 (66.9) | 117 (33.1) |
| Communication between different health care providers | 246 (69.7) | 107 (30.3) |
| Decision making involvement | 152 (43.1) | 201 (56.9) |
| Total counseling received during your health center stay | 201 (56.9) | 152 (43.1) |
| Total care and support provided during labor and delivery time | 221 (62.6) | 132 (37.4) |
| Total care and support provided for newborn baby | 228 (64.6) | 125 (35.4) |

## The total quality of BEmONC services from the perspective of mothers

The total quality was computed by considering all three quality assessment aspects; these are input, process, and outcome. The quality was categorized based on the overall mean and its percentage. The proportion of those who scored greater than or equal to the overall mean of 104 (70%) or rated the quality as good in this study was 201 (56.9%) with a 95% confidence interval (51.6, 62.2). The availability of functional, clean shower and toilet rooms 135 (38.2%), involving them in decision-making about them and their baby's condition 152 (43.1%), and asking permission before applying any examination and procedures 171 (48.4%) were among the most rated as poorly addressed aspects in BEmONC services from the perspective of mothers in this study.

## Characteristics related to the quality of BEmONC services from the perspective of mothers

In a multivariable analysis, mothers aged between 20 and 24 years old had a lower likelihood of viewing the quality of services as good than those > 35 years old (AOR = 0.362; 95% CI: 0.157–0.837). On the other hand, respondents whose average monthly income was less than 1500 ETB had two times higher odds of viewing the quality as good than those with an average monthly income of above 6000 ETB (AOR = 2.429; 95% CI: 1.026–5.753).

The presence of a companion was also one of the most important predictors of quality services; mothers who had a companion throughout their stay had an 18-fold higher chance of rating the quality as good than those alone (AOR = 18.557; 95% CI: 3.844–89.588). With regard to payment, respondents who had paid for any services or products during their stay in the health centers had lower odds of rating the quality as good than those freely serviced (AOR = 0.564; 95% CI: 0.327–0.971) (Table 6).

**Table 6. Factors associated with the quality of BEmONC services from the perspective of mothers in bivariate and multivariable analysis in public health centers of Addis Ababa, Ethiopia, 2022.**

| Factors | Overall Quality | | COR (95% CI) | AOR (95% CI) | P-value |
|---|---|---|---|---|---|
| | Good | Poor | | | |
| **Age category** | | | | | |
| 15–19 | 2 | 1 | 1.226(0.104,14.455) | 0.53(0.042,6.642) | 0.623 |
| 20–24 | 27 | 41 | 0.404(0.191,0.854) | 0.362(0.157,0.837) | **0.017**[*] |
| 25–29 | 93 | 60 | 0.95(0.493,1.832) | 0.826(0.39,1.75) | 0.618 |
| 30–34 | 48 | 31 | 0.949(0.458,1.965) | 0.901(0.393,2.062) | 0.804 |
| >35 | 31 | 19 | 1 | 1 | |
| **Average monthly income category** | | | | | |
| 0-1500 ETB | 27 | 13 | 2.077(0.966,4.464) | 2.429(1.026,5.753) | **0.044**[*] |
| 1501-3000 ETB | 59 | 41 | 1.439(0.827,2.503) | 1.487(0.81,2.728) | 0.2 |
| 3001-4500 ETB | 24 | 19 | 1.263(0.618,2.580) | 1.388(0.646,2.983) | 0.401 |
| 4501-6000 ETB | 39 | 27 | 1.444(0.774,2.694) | 1.352(0.682,2.68) | 0.387 |
| > 6000 ETB | 52 | 52 | 1 | 1 | |
| **Presence of companion during stay** | | | | | |
| Yes | 199 | 134 | 13.336(3.051,58.55) | 18.557(3.844,89.588) | **0.000**[*] |
| No | 2 | 18 | 1 | 1 | |
| **Payment** | | | | | |
| Yes | 41 | 45 | 0.609(0.374,0.993) | 0.564(0.327,0.971) | **0.039**[*] |
| No | 160 | 107 | 1 | 1 | |

## Discussion

The proportion of good quality from the perspective of mothers in this study was 56.9% with a 95% C.I. (51.6, 62.2), which is lower than a study conducted in Irbid, North Jordan, 64% [19]. This disparity may be attributed to variations in healthcare infrastructure, resource availability, study design, assessment methods, and sample size. Additionally, patient expectations, prior healthcare experiences, and the type of health facilities may have influenced the difference.

When compared to studies conducted in Tigray, Ethiopia, the reported quality in this study (56.9%) was lower than that in three zones in Tigray (65.62%) and Adigrat, Tigray (66.3%) [7,20]. Possible reasons for this discrepancy include differences in study population, the types of health facilities, and mothers' expectations of care, which can be influenced by urban versus rural residence, educational status, and income level.

However, the finding in this study showed a higher quality rating than a study done in the Jabi Tehinan district of Northwest Ethiopia, where only 13% of mothers reported good quality of intrapartum care services [21]. This difference may be due to variations in healthcare infrastructure, health workforce capacity, and study setting, as Addis Ababa, being the capital city, has relatively better-equipped health centers and a higher number of skilled health professionals. Differences in study design and period may have also contributed to the observed variation.

In this study, several factors were found to be significantly associated with the quality of BEmONC services. These include maternal age, average monthly income, the presence of companions, and payment for services. This is consistent with a study conducted in Assela Hospital, Arsi, Oromia, Ethiopia, in which age, monthly income, and payment were important predictors [12].

Specifically, mothers aged between 20 and 24 years were less likely to report good quality compared to those above 35 years in this study; this implies that older mothers may have different expectations or experiences that lead to higher satisfaction. However, this finding contrasts with the Assela Hospital study, where women aged between 20 and 34 years reported higher satisfaction than those aged 35–49 years [12].

Income was another significant determinant of the quality of BEmONC from the mothers' perspective. Mothers with an average monthly income below 1,500 ETB had twice the likelihood of rating the quality of care as good compared to those earning above 6,000 ETB. This finding is in line with a study done in Assela Hospital, Arsi, Oromia, Ethiopia [12]. The possible reasons may be due to expectations based on income level, suggesting that lower-income women may have fewer healthcare options and lower expectations, leading to relatively higher satisfaction with public health centers. Higher-income women, by contrast, may have greater access to private healthcare and have higher expectations, which, when unmet, may lead to lower ratings of public health center services.

The presence of a companion was one of the most important predictors of the quality of BEmONC service. Mothers who had a companion or attendant had 18-fold higher odds of rating quality as good compared to those who were alone. This finding is in line with studies from Brazil and Adigrat, Tigray, Ethiopia [7,22]. This may be due to emotional and psychological support provided by the companions and reduced feelings of neglect. In addition, the reason could also be due to the mother's expectations and the way in which they perceived their care.

Additionally, payment for services was a significant predictor of the quality of BEmONC services. Respondents who had paid for the services had lower odds of reporting good quality compared to those who received free services. This finding is consistent with a study done in three hospitals in Amhara, Ethiopia, where mothers who paid less than or equal to one

hundred fifty-seven ETB were more satisfied than those who paid above one hundred fifty-seven ETB [8]. This could be because of the poor socio-economic status of the women.

## Strengths and limitations of the study

The strength of this study is its use of the Donabedian framework to assess quality BEmONC services from the perspective of mothers by using all three aspects of quality assessment methods: input, process, and outcome (satisfaction). This study was multi-centered, which increases the representativeness of the findings. This study used ODK version 2022.1.2 software to collect the data, along with a Kobo Toolbox humanitarian response server to store the collected data, and it minimizes error.

However, limitations include that the data was limited to deliveries in public health centers, restricting generalization to mothers' total childbirth experiences in health facilities. This study, like other cross-sectional studies, has its own set of design limitations. Utilizing only a quantitative approach to assess quality may not be sufficient to see how mothers' experience. The fact that the study was conducted in a health facility may have influenced the results in favor of the healthcare professionals. Additionally, potential recall and response bias, as mothers' satisfaction may have been influenced by their most recent experience rather than an objective assessment of care quality.

## Conclusion

The total quality of BEmONC services from the viewpoint of mothers was suboptimal. This research strongly recommends that more should be done to ensure that the services given are more client-centered. The mothers stated that the most contributing aspects to rating the quality as poor in this study were the inadequate availability of functional, clean shower and toilet rooms; not involving them in decision-making about them as well as their baby's condition; and not asking permission before applying any examination and procedures.

Mothers between the ages of 20 and 24 years, as well as mothers who paid for the services, were more likely to rate the services as poor. On the other hand, mothers who had a companion or attendant during their stay and whose average monthly income was less than 1500 ETB were more likely to rate the services as good quality.

## Supporting information

**S1 File. English version questionnaire to Plos.**
(DOCX)

**S2 File. Quality of BeMONC SPSS.**
(XLSX)

## Author contributions

**Conceptualization:** Willi Bahre.

**Data curation:** Willi Bahre.

**Formal analysis:** Willi Bahre.

**Methodology:** Willi Bahre.

**Software:** Willi Bahre.

**Supervision:** Achamyelesh Tadele, Finot Debebe.

**Writing – original draft:** Willi Bahre.

**Writing – review & editing:** Achamyelesh Tadele, Finot Debebe.

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
