## [Decision Letter · Decision Letter 0]

23 Jan 2025

PONE-D-24-01243Quality of Basic Emergency Obstetric and Newborn Care Services from Patients’ Perspective in Selected Public Health Centers in Addis Ababa, Ethiopia, 2022: A Cross-Sectional StudyPLOS ONE

Dear Dr. Bahre,

Thank you for submitting your manuscript to PLOS ONE. After careful consideration, we feel that it has merit but does not fully meet PLOS ONE’s publication criteria as it currently stands. Therefore, we invite you to submit a revised version of the manuscript that addresses the points raised during the review process.

We look forward to receiving your revised manuscript.

Kind regards,

Asaye, PhD 

Academic Editor

PLOS ONE

Journal Requirements:

**Additional Editor Comments:**

Reviewers 1

1) Summary of the research and your overall impression

This was a research conducted to assess the quality of Basic Emergency Obstetric and New-born Care services from patients' perspective in 11 public Health Facilities in Addis Ababa Ethiopia. The paper sought to establish the contribution of mothers' perspective on the care they received at the health facility in relation to their health outcomes. It provided evidence that factors such as patient purchasing power ,the patient being accompanied to the hospital and the mothers' age influenced their rating of the services received from the health facility. This study is relevant to the current literature because it allows policy makers to understand challenges which when addressed can increase the quality of care an overall reduce maternal and new-born mortality. Also, the study provides a unique opportunity to understand behavioural challenges that negatively impact on the experience of mothers while at the health facility such as administration of medication without informing them or not being courteous toward the mothers. Furthermore the study was well researched with current and relevant literature.

However, I feel the information on the quality of BEmONC based on the BEmONC signal functions is lacking which will be insightful, particularly in their appreciation of whether the dispensation of all signal functions from the mother's perspective aligns with what is expected as per the national guidelines. Also, in line 4 of page 6 the author states that the health facilities were selected through a simple random sampling method and does not clarify whether all eleven health facilities provided BEmONC services. Overall this was a well researched subject that has the potential to influence policies in the domain of preventing and managing obstetrical complications.

2)Major Issues

• It is not clear from the write up how many of the chosen facilities actually practiced BEmONC. I suggest the author states in the form of a table the facilities and categorize them as performing BEmONC or not, in addition the catchment populations of the facilities should be included to allow readers have an idea of the patient influx ,finally the number of patient captured per facility should be stated as this allows readers to better understand the conclusions drawn from the study.

• The author demonstrated a good mastery of the subject in the discussion section by pointing out discrepancies and using other studies to support their claims ,however the discussion was not well structured alternatively, the author should use the STROBE guideline linked here:https://www.strobe-statement.org/ to improve the structure of the discussion section.

• The authors clearly put in efforts to ensure their study was conformed to ethical considerations. However in paragraph 2 of page 9 entitled "Ethical Consideration" ,the author states in line 7 that verbal consent was obtained from participants ,however we note on table 1 the age group includes minors of 15 years ,could the author state if consent was obtained from their guardians for the purpose of this study.

• Under methods and materials of page 5, the author failed to indicate who was included or excluded from the study even though they belonged to the study population, to avoid confusion the author should boldly outline and inclusion and exclusion criteria for the study.

• In the discussion section the author aimed to demonstrate that discrepancies observed between the studies compared could be due to element such as a difference in culture , difference in national guideline.... however this approach undermines the efforts of the team by not comparing valuable information such as the difference in study design, study population, or the methodology used in both studies. In case the author wishes to attribute the differences to the culture, or the national guideline or other element as they may see fit , this should be substantiated by stating exactly what the difference is.

3) Minor Issues

• In paragraph 3 of the discussion entitled study limitation the author states element that reduce the validity of the study and failed to demonstrate the strengths of the study that that minimized the study limitations. Alternatively, the author should state the study limitations and how it was minimized.

• On page 7 line 15 ,the definition of signal function, "Consequences" should be replaced by "Complications"

• On Page 7 line 17 "is" should be removed.

Reviewer 2

Comments to the Author

Manuscript Number: PONE-D-24-01243

The manuscript titled " Quality of Basic Emergency Obstetric and Newborn Care Services from Patients’ Perspective in Selected Public Health Centers in Addis Ababa, Ethiopia, 2022: A Cross-Sectional Study " is important research that adds to present knowledge on obstetric violence. The study was generally well conducted but the authors need to address pertinent issues.

My comments are as follows:

Abstract:

Line 24: The gap in this study has not been adequately addressed, highlighting a significant oversight in the research process. Without a thorough exploration of this gap, the study may lack depth and fail to contribute meaningfully to the existing body of knowledge. So, addressing this issue will enhance the overall rigor and relevance of the research.

Next to line 27: Please ensure that the aim/objective of the study are included into the abstract section. This addition will provide clarity and context for readers.

Line 34: The final model identified several significant variables: age, income, companion, and payment. However, the results section only addressed two of these variables—companion and payment. For clarity and consistency, all significant variables should be discussed.

Line 42: Add your recommendation

Background

The discussion is well-presented, but addressing the gaps will strengthen it and make it more convincing.

Methods and material

Study setting and period

I recommend that you include the number of health facilities, the number of healthcare providers—particularly obstetric healthcare providers—and the number of women who utilized BEmONC services.

Population

How did you determine whether the women received at least one Basic Emergency Obstetric and Newborn Care (BEmONC) signal function service in order to include them in your study?

Line 121-123: Please provide the average number of one-month postnatal mothers to clarify the calculation of the k-th interval for selecting the study population.

Operational definition

The author should provide clear definitions for all components of the Donabedian model(input, process and output) to offer a more detailed understanding.

Results

Socio demographic features

Line 229: Since the table includes economic variables, it would be more appropriate to refer to it as sociodemographic/economic features.

Line 231: What were the reasons for the 6% of your sample size that did not participate in the study?

Table 1: You have three participants whose ages are under 18, classifying them as minors. This population typically requires support from their guardians or partners to participate in the study. Did you obtain assent from these minors before conducting the interviews? Additionally, how did you address the ethical considerations related to including children in your research activities?

Table 1: While the study thoroughly examined all variables related to women's sociodemographic, it notably overlooked the variables associated with husbands or partners. This omission is significant, as the role of a husband can greatly influence various aspects of women's health and well-being.

Table 1: ` What was the rationale behind classifying income in this manner? The classification of women's income is unclear. Could you please provide a detailed explanation of how it was operationalized?

Obstetric characteristics of participants

Line 272: I recommend creating a table to present the obstetric characteristics, as this will help readers easily grasp the overall context of maternal-related issues.

Line 2273-274: While it is acceptable to report whether the mothers had antenatal care (ANC) visits, what is crucial is the total number of contacts these women had with healthcare providers. This aspect has not been addressed in the study.

Line 274: “339 (96%) of them wanted pregnancy for this current one” …. This not clear. Please make it clary and understandable by the readers

Line 278-280: Do you think women can determine properly the amount of time they spend before receiving the services. Have you cross checked their responses with other mechanisms

Line 280: Companionship is an important intervention that enhances maternal healthcare services, and it is also a fundamental right for women to have during the utilization of maternal health services. In the study, “Twenty (5.7%) of the women had no companion (attendant) 281 during their stay “…this is too general. So, it should be specified where (service units) women utilize the companionship.

Line 285-286: The study revealed “Two-thirds (75.6%) of the 286 women did not pay for any of the services in the health centers during their stay.” This indicates that approximately 25% of women have covered their medical expenses. However, the Ethiopian government has exempted nearly all maternal healthcare services from charges for women. What accounts for this discrepancy in your findings?

Line 289- 290: in the methods section you operationalized good quality as the quality of BEmONC services from the mothers’ perspective when the mothers 161 scored greater than or equal to the mean of 104 (70%) of all the quality questions. However; in the result section good quality defined as those who scored greater than or equal to the mean of 15 (60%) or rated as good quality, was 174 291 (49.3%). This is contradicted with each other. Look at also the quality measures for process and output. Please see it carefully.

Line 377: In addition to the percentage, please include the frequency for the variables. For example, you could present it as frequency (56.9%).

Table 5: Avoid relying solely on statistical hypothesis testing, such as P values, which fail to convey important information about effect size and precision of estimates. thus, focusing on confidence interval can clearly explain the significance of the variables. Better if you delete the p-value.

Table 5: I have observed wide confidence interval in the final model. A wide confidence interval (CI) indicates a high level of uncertainty regarding the estimated parameter, suggesting that the sample does not provide a precise representation of the population. It suggests that there is insufficient evidence to make definitive conclusions about the parameter being estimated. This can occur due to smaller sample size, high variability or dispersion within the sample data and others. check it.

Discussion

Line 435- 437: the way you justify somehow it is not convincing. Please discuss how your findings relate to existing literature, highlighting both similarities and differences with previous studies.

The study compares and contrasts only a limited number of variables and lacks evidence-based justifications for the discrepancies and similarities between the studies. Therefore, the author should provide strong scientific reasoning for any inline studies and those that align with the findings.

Limitation

While it is important to acknowledge the limitations of your study, it is equally essential to discuss its strengths. Why was this aspect overlooked in your explanation?

Conclusion

Line 473 says the total quality of BEmONC services from the viewpoint of mothers was suboptimal. What was your criteria to declare as it is suboptimal? What was your comparator?

Reviewers' comments:

Reviewer's Responses to Questions

**Comments to the Author**

1. Is the manuscript technically sound, and do the data support the conclusions?

Reviewer #1: Yes

Reviewer #2: Yes

2. Has the statistical analysis been performed appropriately and rigorously? 

Reviewer #1: Yes

Reviewer #2: Yes

3. Have the authors made all data underlying the findings in their manuscript fully available?

Reviewer #1: Yes

Reviewer #2: Yes

4. Is the manuscript presented in an intelligible fashion and written in standard English?

Reviewer #1: No

Reviewer #2: Yes

5. Review Comments to the Author

Reviewer #1: 1) Summary of the research and your overall impression

This was a research conducted to assess the quality of Basic Emergency Obstetric and New-born Care services from patients' perspective in 11 public Health Facilities in Addis Ababa Ethiopia. The paper sought to establish the contribution of mothers' perspective on the care they received at the health facility in relation to their health outcomes. It provided evidence that factors such as patient purchasing power ,the patient being accompanied to the hospital and the mothers' age influenced their rating of the services received from the health facility. This study is relevant to the current literature because it allows policy makers to understand challenges which when addressed can increase the quality of care an overall reduce maternal and new-born mortality. Also, the study provides a unique opportunity to understand behavioural challenges that negatively impact on the experience of mothers while at the health facility such as administration of medication without informing them or not being courteous toward the mothers. Furthermore the study was well researched with current and relevant literature.

However, I feel the information on the quality of BEmONC based on the BEmONC signal functions is lacking which will be insightful, particularly in their appreciation of whether the dispensation of all signal functions from the mother's perspective aligns with what is expected as per the national guidelines. Also, in line 4 of page 6 the author states that the health facilities were selected through a simple random sampling method and does not clarify whether all eleven health facilities provided BEmONC services. Overall this was a well researched subject that has the potential to influence policies in the domain of preventing and managing obstetrical complications.

2)Major Issues

• It is not clear from the write up how many of the chosen facilities actually practiced BEmONC. I suggest the author states in the form of a table the facilities and categorize them as performing BEmONC or not, in addition the catchment populations of the facilities should be included to allow readers have an idea of the patient influx ,finally the number of patient captured per facility should be stated as this allows readers to better understand the conclusions drawn from the study.

• The author demonstrated a good mastery of the subject in the discussion section by pointing out discrepancies and using other studies to support their claims ,however the discussion was not well structured alternatively, the author should use the STROBE guideline linked here:https://www.strobe-statement.org/ to improve the structure of the discussion section.

• The authors clearly put in efforts to ensure their study was conformed to ethical considerations. However in paragraph 2 of page 9 entitled "Ethical Consideration" ,the author states in line 7 that verbal consent was obtained from participants ,however we note on table 1 the age group includes minors of 15 years ,could the author state if consent was obtained from their guardians for the purpose of this study.

• Under methods and materials of page 5, the author failed to indicate who was included or excluded from the study even though they belonged to the study population, to avoid confusion the author should boldly outline and inclusion and exclusion criteria for the study.

• In the discussion section the author aimed to demonstrate that discrepancies observed between the studies compared could be due to element such as a difference in culture , difference in national guideline.... however this approach undermines the efforts of the team by not comparing valuable information such as the difference in study design, study population, or the methodology used in both studies. In case the author wishes to attribute the differences to the culture, or the national guideline or other element as they may see fit , this should be substantiated by stating exactly what the difference is.

3) Minor Issues

• In paragraph 3 of the discussion entitled study limitation the author states element that reduce the validity of the study and failed to demonstrate the strengths of the study that that minimized the study limitations. Alternatively, the author should state the study limitations and how it was minimized.

• On page 7 line 15 ,the definition of signal function, "Consequences" should be replaced by "Complications"

• On Page 7 line 17 "is" should be removed.

Reviewer #2: Comments to the Author

Manuscript Number: PONE-D-24-01243

The manuscript titled " Quality of Basic Emergency Obstetric and Newborn Care Services from Patients’ Perspective in Selected Public Health Centers in Addis Ababa, Ethiopia, 2022: A Cross-Sectional Study " is important research that adds to present knowledge on obstetric violence. The study was generally well conducted but the authors need to address pertinent issues.

My comments are as follows:

Abstract:

Line 24: The gap in this study has not been adequately addressed, highlighting a significant oversight in the research process. Without a thorough exploration of this gap, the study may lack depth and fail to contribute meaningfully to the existing body of knowledge. So, addressing this issue will enhance the overall rigor and relevance of the research.

Next to line 27: Please ensure that the aim/objective of the study are included into the abstract section. This addition will provide clarity and context for readers.

Line 34: The final model identified several significant variables: age, income, companion, and payment. However, the results section only addressed two of these variables—companion and payment. For clarity and consistency, all significant variables should be discussed.

Line 42: Add your recommendation

Background

The discussion is well-presented, but addressing the gaps will strengthen it and make it more convincing.

Methods and material

Study setting and period

I recommend that you include the number of health facilities, the number of healthcare providers—particularly obstetric healthcare providers—and the number of women who utilized BEmONC services.

Population

How did you determine whether the women received at least one Basic Emergency Obstetric and Newborn Care (BEmONC) signal function service in order to include them in your study?

Line 121-123: Please provide the average number of one-month postnatal mothers to clarify the calculation of the k-th interval for selecting the study population.

Operational definition

The author should provide clear definitions for all components of the Donabedian model(input, process and output) to offer a more detailed understanding.

Results

Socio demographic features

Line 229: Since the table includes economic variables, it would be more appropriate to refer to it as sociodemographic/economic features.

Line 231: What were the reasons for the 6% of your sample size that did not participate in the study?

Table 1: You have three participants whose ages are under 18, classifying them as minors. This population typically requires support from their guardians or partners to participate in the study. Did you obtain assent from these minors before conducting the interviews? Additionally, how did you address the ethical considerations related to including children in your research activities?

Table 1: While the study thoroughly examined all variables related to women's sociodemographic, it notably overlooked the variables associated with husbands or partners. This omission is significant, as the role of a husband can greatly influence various aspects of women's health and well-being.

Table 1: ` What was the rationale behind classifying income in this manner? The classification of women's income is unclear. Could you please provide a detailed explanation of how it was operationalized?

Obstetric characteristics of participants

Line 272: I recommend creating a table to present the obstetric characteristics, as this will help readers easily grasp the overall context of maternal-related issues.

Line 2273-274: While it is acceptable to report whether the mothers had antenatal care (ANC) visits, what is crucial is the total number of contacts these women had with healthcare providers. This aspect has not been addressed in the study.

Line 274: “339 (96%) of them wanted pregnancy for this current one” …. This not clear. Please make it clary and understandable by the readers

Line 278-280: Do you think women can determine properly the amount of time they spend before receiving the services. Have you cross checked their responses with other mechanisms

Line 280: Companionship is an important intervention that enhances maternal healthcare services, and it is also a fundamental right for women to have during the utilization of maternal health services. In the study, “Twenty (5.7%) of the women had no companion (attendant) 281 during their stay “…this is too general. So, it should be specified where (service units) women utilize the companionship.

Line 285-286: The study revealed “Two-thirds (75.6%) of the 286 women did not pay for any of the services in the health centers during their stay.” This indicates that approximately 25% of women have covered their medical expenses. However, the Ethiopian government has exempted nearly all maternal healthcare services from charges for women. What accounts for this discrepancy in your findings?

Line 289- 290: in the methods section you operationalized good quality as the quality of BEmONC services from the mothers’ perspective when the mothers 161 scored greater than or equal to the mean of 104 (70%) of all the quality questions. However; in the result section good quality defined as those who scored greater than or equal to the mean of 15 (60%) or rated as good quality, was 174 291 (49.3%). This is contradicted with each other. Look at also the quality measures for process and output. Please see it carefully.

Line 377: In addition to the percentage, please include the frequency for the variables. For example, you could present it as frequency (56.9%).

Table 5: Avoid relying solely on statistical hypothesis testing, such as P values, which fail to convey important information about effect size and precision of estimates. thus, focusing on confidence interval can clearly explain the significance of the variables. Better if you delete the p-value.

Table 5: I have observed wide confidence interval in the final model. A wide confidence interval (CI) indicates a high level of uncertainty regarding the estimated parameter, suggesting that the sample does not provide a precise representation of the population. It suggests that there is insufficient evidence to make definitive conclusions about the parameter being estimated. This can occur due to smaller sample size, high variability or dispersion within the sample data and others. check it.

Discussion

Line 435- 437: the way you justify somehow it is not convincing. Please discuss how your findings relate to existing literature, highlighting both similarities and differences with previous studies.

The study compares and contrasts only a limited number of variables and lacks evidence-based justifications for the discrepancies and similarities between the studies. Therefore, the author should provide strong scientific reasoning for any inline studies and those that align with the findings.

Limitation

While it is important to acknowledge the limitations of your study, it is equally essential to discuss its strengths. Why was this aspect overlooked in your explanation?

Conclusion

Line 473 says the total quality of BEmONC services from the viewpoint of mothers was suboptimal. What was your criteria to declare as it is suboptimal? What was your comparator?

6. PLOS authors have the option to publish the peer review history of their article (what does this mean?). If published, this will include your full peer review and any attached files.

Reviewer #1: No

Reviewer #2: No

---

## [Author Response · Author response to Decision Letter 1]

9 Feb 2025

January 28, 2025

Dear Editor,

Thank you for giving us the opportunity to submit the revised draft of our manuscript entitled “Quality of Basic Emergency Obstetric and Newborn Care Services from Patients’ Perspective in Selected Public Health Centers in Addis Ababa, Ethiopia 2022: A Cross-Sectional Study” to PLOS ONE. We appreciate the time and effort that you and the reviewers have dedicated to providing your valuable feedback on our manuscript. We are grateful to you and the reviewers for the insightful comments on our paper. We have tried to revise our manuscript in accordance with the suggestions and comments provided by you and the reviewers.

Here is a point-by-point response to the comments made by the reviewers, the editor, and the editorial staff.

Sincerely,

Willi Bahre

Adigrat University, Ethiopia

E-mail: willibahre21@gmail.com

Phone number: +251-9 77755583

Part One: Point-by-point responses to editor and editorial staff

A rebuttal letter that responds to each point raised by the academic editor and reviewer(s). You should upload this letter as a separate file labeled 'Response to Reviewers.

Response: Uploaded

A marked-up copy of your manuscript that highlights changes made to the original version. You should upload this as a separate file labeled 'Revised Manuscript with Track Changes.

Response: A revised manuscript with track changes is submitted in accordance with the instruction.

An unmarked version of your revised paper without tracked changes. You should upload this as a separate file labeled 'Manuscript.

Response: A revised manuscript without tracked changes is submitted in accordance with the instruction.

Response: It meets PLOS ONE’s style requirements.

2. We suggest you thoroughly copyedit your manuscript for language usage, spelling, and grammar.

Response: Modified

Response: Incorporated

4. Please include captions for your Supporting Information files at the end of your manuscript, and update any in-text citations to match accordingly.

Response: Modified

Part Two: Point-by-point responses to reviewers

Reviewer 1

Thank you, dear reviewer, for reviewing our paper. We have answered each of your points below.

1. However, I feel the information on the quality of BEmONC based on the BEmONC signal functions is lacking which will be insightful, particularly in their appreciation of whether the dispensation of all signal functions from the mother's perspective aligns with what is expected as per the national guidelines.

Response: Dear reviewer, thank you for your valuable feedback. We fully agree with your idea. In response, we have incorporated all seven signal functions of Basic Emergency Obstetric and Newborn Care (BEmONC) services and analyzed them from the mother’s perspective to determine whether their delivery aligns with what is stipulated by the national guidelines.

Also, in line 4 of page 6 the author states that the health facilities were selected through a simple random sampling method and does not clarify whether all eleven health facilities provided BEmONC services.

Response: We agreed with this comment, dear reviewer. Yes, all the selected eleven public health centers in Addis Ababa provided Basic Emergency Obstetric and Newborn Care (BEmONC) services. Therefore, we have incorporated it in the revised manuscript.

2. Major Issues

It is not clear from the write up how many of the chosen facilities actually practiced BEmONC. I suggest the author states in the form of a table the facilities and categorize them as performing BEmONC or not, in addition the catchment populations of the facilities should be included to allow readers have an idea of the patient influx, finally the number of patients captured per facility should be stated as this allows readers to better understand the conclusions drawn from the study.

Response: Dear Reviewer, Thank you for your thoughtful comment. To clarify, all the selected eleven public health centers in Addis Ababa, Ethiopia practiced Basic Emergency Obstetric and Newborn Care (BEmONC) services. Since our study was aimed at assessing the quality of BEmONC from a mother's perspective rather than the availability of BEmONC services, it is not necessary to create a table. Specifically, we have included the catchment populations of the selected facilities to provide insight into patient influx. Additionally, we have stated the number of patients captured per facility. We have now made this explicit in the revised manuscript to ensure clarity.

The author demonstrated a good mastery of the subject in the discussion section by pointing out discrepancies and using other studies to support their claims, however the discussion was not well structured alternatively, the author should use the STROBE guideline linked here: https://www.strobe-statement.org/ to improve the structure of the discussion section.

Response: We agreed with this comment, dear reviewer. Modified using STROBE guideline in the revised manuscript.

The authors clearly put in efforts to ensure their study was conformed to ethical considerations. However, in paragraph 2 of page 9 entitled "Ethical Consideration”, the author states in line 7 that verbal consent was obtained from participants, however we note on table 1 the age group includes minors of 15 years, could the author state if consent was obtained from their guardians for the purpose of this study.

Response: Dear Reviewer, thank you for your insightful comment. We fully accept this suggestion. While there were three participants in the age category of 15–19 years, it is important to note that two of them were 18 years old, and the third was 19 years old. Therefore, verbal consent was obtained directly from the participants themselves. We appreciate your feedback and the opportunity to clarify this aspect of our study.

Under methods and materials of page 5, the author failed to indicate who was included or excluded from the study even though they belonged to the study population, to avoid confusion the author should boldly outline and inclusion and exclusion criteria for the study.

Response: We appreciate your valuable comment and fully agree with your suggestion. In response, we have explicitly defined the inclusion and exclusion criteria in the revised manuscript to ensure clarity and a better understanding. (Line 116-122)

In the discussion section the author aimed to demonstrate that discrepancies observed between the studies compared could be due to element such as a difference in culture, difference in national guideline.... however, this approach undermines the efforts of the team by not comparing valuable information such as the difference in study design, study population, or the methodology used in both studies.

Response: Dear reviewer, thank you for your valuable comment. We modified it in the revised document.

3. Minor Issues

In paragraph 3 of the discussion entitled study limitation the author states element that reduce the validity of the study and failed to demonstrate the strengths of the study that that minimized the study limitations. Alternatively, the author should state the study limitations and how it was minimized.

Response: Modified (Line 501-506)

On page 7 line 15, the definition of signal function, "Consequences" should be replaced by "Complications"

Response: Changed (Line 163)

On Page 7 line 17 "is" should be removed.

Response: Removed (Line 165)

Reviewer 2

Thank you, dear reviewer, for reviewing our paper. We have answered each of your points below.

1. Abstract

Line 24: The gap in this study has not been adequately addressed, highlighting a significant oversight in the research process. Without a thorough exploration of this gap, the study may lack depth and fail to contribute meaningfully to the existing body of knowledge. So, addressing this issue will enhance the overall rigor and relevance of the research.

Response: Thank you in advance dear for your critical reviewing. We modified it the revised document (Line 27-29).

Next to line 27: Please ensure that the aim/objective of the study are included into the abstract section. This addition will provide clarity and context for readers.

Response: We definitely agreed with your comment dear. Modified in the revised manuscript (Line 30-32).

Line 34: The final model identified several significant variables: age, income, companion, and payment. However, the results section only addressed two of these variables companion and payment. For clarity and consistency, all significant variables should be discussed.

Response: Modified (Line 42-44 and 46-48)

Line 42: Add your recommendation

Response: Modified (Line 51 and 52)

2. Background

The discussion is well-presented, but addressing the gaps will strengthen it and make it more convincing.

Response: We accepted the comment and modified.

3. Methods and material

Study setting and period

I recommend that you include the number of health facilities, the number of healthcare providers, particularly obstetric healthcare providers and the number of women who utilized BEmONC services.

Response: Modified (Line 101-103, 105-108)

Population

How did you determine whether the women received at least one Basic Emergency Obstetric and Newborn Care (BEmONC) signal function service in order to include them in your study?

Response: We know by asking postpartum women about the interventions they received during labor and delivery.

Line 121-123: Please provide the average number of one-month postnatal mothers to clarify the calculation of the kth interval for selecting the study population.

Response: Modified (Line 134-137)

Operational definition

The author should provide clear definitions for all components of the Donabedian model (input, process and output) to offer a more detailed understanding.

Response: We agreed with your comment and we have added additional operational definitions that we have used. (Line 184-188)

4. Results

Socio demographic features

Line 229: Since the table includes economic variables, it would be more appropriate to refer to it as sociodemographic/economic features.

Response: Incorporated in the revised manuscript. (Line 236)

Line 231: What were the reasons for the 6% of your sample size that did not participate in the study?

Response: 24 (6%) of the participants were not volunteer to participate (they refused to take part in the interview). So, this was compensated by non-response rate. Since we have considered 10% non-response rate during sample size calculation.

Table 1: You have three participants whose ages are under 18, classifying them as minors. This population typically requires support from their guardians or partners to participate in the study. Did you obtain assent from these minors before conducting the interviews? Additionally, how did you address the ethical considerations related to including children in your research activities?

Response: Dear reviewer, thank you in advance for your critical reviewing. We fully accept this suggestion. While there were three participants in the age category of 15–19 years, it is important to note that two of them were 18 years old, and the third was 19 years old. Therefore, since we have no minors in our study verbal consent was obtained directly from the participants themselves.

Table 1: While the study thoroughly examined all variables related to women's sociodemographic, it notably overlooked the variables associated with husbands or partners. This omission is significant, as the role of a husband can greatly influence various aspects of women's health and well-being.

Response: We appreciate your concern dear. However, we are incorporated important husband related variables such as husband’s education, husband’s occupation in table one and presence of companion or attendant in the obstetric characteristics of participants part.

Table 1: What was the rationale behind classifying income in this manner? The classification of women's income is unclear. Could you please provide a detailed explanation of how it was operationalized?

Response: Dear Reviewer, Thank you for your comment. We have taken this category of women’s income from a study conducted in Adigrat, Tigray, Ethiopia, which focused on a similar topic. We believe this source is relevant and aligns with the context of our data.

Obstetric characteristics of participants

Line 272: I recommend creating a table to present the obstetric characteristics, as this will help readers easily grasp the overall context of maternal-related issues.

Response: Modified it in the revised document. (Line 295)

Line 273-274: While it is acceptable to report whether the mothers had antenatal care (ANC) visits, what is crucial is the total number of contacts these women had with healthcare providers. This aspect has not been addressed in the study.

Response: We definitely agreed with your comment dear. Our intention was to assess the quality of BEmONC services and associated factors from the perspective of mothers in selected public health centers in Addis Ababa, Ethiopia. So, we think it is a sufficient indicator to report the presence of ANC follow-up rather than the number of ANC visits.

Line 274: “339 (96%) of them wanted pregnancy for this current one” …. This not clear. Please make it clary and understandable by the readers.

Response: Clarified in the revised manuscript. (Line 275 and 276)

Line 278-280: Do you think women can determine properly the amount of time they spend before receiving the services. Have you cross checked their responses with other mechanisms.

Response: We definitely agreed with your comment dear. Yes, we have cross checked it with their medical records.

Line 280: Companionship is an important intervention that enhances maternal healthcare services, and it is also a fundamental right for women to have during the utilization of maternal health services. In the study, “Twenty (5.7%) of the women had no companion (attendant) during their stay “…this is too general. So, it should be specified where (service units) women utilize the companionship.

Response: Dear reviewer, we accepted this comment. Modified (Line 279 and 280)

Line 285-286: The study revealed “Two-thirds (75.6%) of the women did not pay for any of the services in the health centers during their stay.” This indicates that approximately 25% of women have covered their medical expenses. However, the Ethiopian government has exempted nearly all maternal healthcare services from charges for women. What accounts for this discrepancy in your findings?

Response: Dear reviewer, thank you for your valuable feedback. Even though, the Ethiopian government has exempted nearly all maternal healthcare services are free from charges; in our study 24.4% of the women reported that they went to nearby private clinics for different laboratory checkups and ultrasound due to unavailability of the services in the health centers during their stay.

Line 289- 290: In the methods section you operationalized good quality as the quality of BEmONC services from the mothers’ perspective when the mothers scored greater than or equal to the mean of 104 (70%) of all the quality questions. However; in the result section good quality defined as those who scored greater than or equal to the mean of 15 (60%) or rated as good quality, was 174 291 (49.3%). This is contradicted with each other. Look at also the quality measures for process and output. Please see it carefully.

Response: Dear reviewer, we accepted this comment. The total good quality from the mothers’ perspective regarding input was, these scored ≥ to the mean of 15 (60), which was 174 (49.3%). The total of good quality from the perspective of women regarding process was, these scored ≥ to the mean of 57 (70%), which was 216 (61.2%). The total of good quality regarding output / satisf

---

## [Decision Letter · Decision Letter 1]

25 Feb 2025

Quality of Basic Emergency Obstetric and Newborn Care Services from Patients’ Perspective in Selected Public Health Centers in Addis Ababa, Ethiopia, 2022: A Cross-Sectional Study

PONE-D-24-01243R1

Dear Dr. Willi_Bahre,

We’re pleased to inform you that your manuscript has been judged scientifically suitable for publication and will be formally accepted for publication once it meets all outstanding technical requirements.

Kind regards,

 Asaye, PhD 

Academic Editor

PLOS ONE

Additional Editor Comments (optional):

Reviewers' comments:

Reviewer's Responses to Questions

**Comments to the Author**

1. If the authors have adequately addressed your comments raised in a previous round of review and you feel that this manuscript is now acceptable for publication, you may indicate that here to bypass the “Comments to the Author” section, enter your conflict of interest statement in the “Confidential to Editor” section, and submit your "Accept" recommendation.

Reviewer #1: All comments have been addressed

Reviewer #2: All comments have been addressed

2. Is the manuscript technically sound, and do the data support the conclusions?

Reviewer #1: Yes

Reviewer #2: Yes

3. Has the statistical analysis been performed appropriately and rigorously? 

Reviewer #1: Yes

Reviewer #2: Yes

4. Have the authors made all data underlying the findings in their manuscript fully available?

Reviewer #1: Yes

Reviewer #2: Yes

5. Is the manuscript presented in an intelligible fashion and written in standard English?

Reviewer #1: Yes

Reviewer #2: Yes

6. Review Comments to the Author

Reviewer #1: (No Response)

Reviewer #2: The author has addressed all comments and suggestions related to the introduction, methodology, data analysis, discussion, and conclusion of the study.

7. PLOS authors have the option to publish the peer review history of their article (what does this mean?). If published, this will include your full peer review and any attached files.

Reviewer #1: No

Reviewer #2: No

---

## [Editor Report · Acceptance letter]

PONE-D-24-01243R1

PLOS ONE

Dear Dr. Bahre,

I'm pleased to inform you that your manuscript has been deemed suitable for publication in PLOS ONE. Congratulations! Your manuscript is now being handed over to our production team.

Kind regards,

on behalf of

Dr. Mengstu Melkamu Asaye

Academic Editor

PLOS ONE